# CRISPR-Cas9-based mutagenesis frequently provokes on-target mRNA misregulation

Rubina Tuladhar [1], Yunku Yeu[2], John Tyler Piazza[1], Zhen Tan[3], Jean Rene Clemenceau[2], Xiaofeng Wu[1], Quinn Barrett[1], Jeremiah Herbert[1], David H. Mathews [3], James Kim [4,5], Tae Hyun Hwang [2] & Lawrence Lum [1]

The introduction of insertion-deletions (INDELs) by non-homologous end-joining (NHEJ) pathway underlies the mechanistic basis of CRISPR-Cas9-directed genome editing. Selective gene ablation using CRISPR-Cas9 is achieved by installation of a premature termination codon (PTC) from a frameshift-inducing INDEL that elicits nonsense-mediated decay (NMD) of the mutant mRNA. Here, by examining the mRNA and protein products of CRISPR targeted genes in a cell line panel with presumed gene knockouts, we detect the production of foreign mRNAs or proteins in ~50% of the cell lines. We demonstrate that these aberrant protein products stem from the introduction of INDELs that promote internal ribosomal entry, convert pseudo-mRNAs (alternatively spliced mRNAs with a PTC) into protein encoding molecules, or induce exon skipping by disruption of exon splicing enhancers (ESEs). Our results reveal challenges to manipulating gene expression outcomes using INDEL-based mutagenesis and strategies useful in mitigating their impact on intended genome-editing outcomes.

[1] Department of Cell Biology, University of Texas Southwestern Medical Center, Dallas, TX 75390, USA. [2] Department of Quantitative Health Sciences, Cleveland Clinic Lerner Research Institute, Cleveland, OH 44195, USA. [3] Department of Biochemistry and Biophysics, University of Rochester Medical Center, Rochester, NY 14642, USA. [4] Department of Internal Medicine, University of Texas Southwestern Medical Center, Dallas, TX 75390, USA. [5] Hamon Center for Therapeutic Oncology Research Department of Biochemistry, University of Texas Southwestern Medical Center, Dallas, TX 75390, USA. Correspondence and requests for materials should be addressed to T.H.H. (email: hwangt@ccf.org) or to L.L. (email: lawrence.lum@utsouthwestern.edu)

Technologies enabling the directed introduction of double-stranded DNA breaks such as CRISPR-Cas9 have transformed our ability to systematically identify DNA sequences important in biology[1,2]. The repair of these double-stranded breaks by non-homologous end-joining (NHEJ) results in insertion-deletions (INDELs) of unpredictable length that upon introduction into exonic sequences could alter the coding frame and install a premature termination codon (PTC). Ribosomes that encounter a PTC in nascent mRNAs, recognized by the assembly of a complex that includes proteins from the ribosome and a 3′ exon–splice junction complex, induces the destruction of the mutant mRNA[3,4]. On the other hand, INDELs that preserve the reading frame may yield proteins with altered sequences and thus shed light on determinants important for its function[5].

Exonic sequences are laden with regulatory features that control many facets of the mRNA lifecycle including splicing and folding, two mRNA attributes that influence protein sequence composition and sites of initiation/termination, respectively[6–8]. Yet, the frequency with which these elements once impacted by INDELs influence gene expression outcomes remains mostly unknown. Another potential obstacle to precision gene editing using INDEL-type mutagenesis is the presence of pseudo-mRNAs, mRNAs harboring a PTC that can nevertheless incorporate introduced INDELs thus altering their potential to produce proteins[9].

To determine the extent to which these molecular events confound our ability to predict gene expression outcomes from CRISPR-Cas9 editing, we have taken inventory of the post-transcriptional and -translational effects of frameshift-inducing INDELs in a panel of CRISPR-edited cells lines. We observe changes in the array of transcripts or proteins expressed from CRISPR-targeted genes in ~50% of the cell lines studied. A mechanistic account of these phenomena is presented here.

## Results

**Unanticipated gene expression outcomes with CRISPR editing.**
To service several ongoing research programs, we had assembled a panel of commercially available HAP1 cell lines harboring frameshift-inducing INDELs that presumably eliminate effective protein production from the targeted gene by promoting nonsense-mediated decay (NMD) of the encoded mRNA (Fig. 1a; Supplementary Table 1). HAP1 cells harbor a single copy of each chromosome thus reducing the challenges frequently associated with achieving homozygosity in diploid cells for genetic studies[10]. To confirm the effects of the INDEL on-target gene expression, we used two antibodies each recognizing a different epitope within the targeted protein (Fig. 1b; Supplementary Table 2). We observed in some cell lines the anticipated loss of protein presumably due to the introduced INDEL but in other instances the appearance of novel proteins detectable by western blot analysis using a single or both antibodies (4/13 cell lines or ~30%; Fig. 1b). For example, in the case of the *TOP1, SIRT1, CTNNB1*, and *LRP6* knockout cell lines, we observed the substitution of the canonical protein for a faster migrating novel protein detected by western blot analysis.

Given our inability to account for the emergence of these novel proteins based on the annotated genetic alteration introduced by CRISPR-Cas9, we next examined the effects of the INDEL on mRNA splicing given that exonic sequences harbor splicing regulatory elements[8,11,12] (Fig. 1c; Supplementary Table 3). In the case of the *TOP1* knockout cell line where we had observed the appearance of a novel TOP1 protein, we also witnessed the emergence of a novel mRNA species. Sequencing a cDNA-derived amplicon from the novel splice variant revealed the

absence of the INDEL-containing exon suggesting the mutant protein was generated by an INDEL-induced exon exclusion event (Supplementary Data 1). In addition to the use of two different antibodies to evaluate TOP1 protein in the CRISPR-edited cell line (Fig. 1b), we also observed enrichment of both the wt and truncated TOP1 protein in the nucleus where the protein is predominantly localized[13] (Fig. 2a). The truncated TOP1 protein nevertheless retained catalytic activity as measured using an enzymatic assay for monitoring relaxation of supercoiled DNA (Fig. 2b). The retention of catalytic activity by the truncated TOP1 protein is consistent with the designation of *TOP1* as an essential gene in HAP1 cells from a gene trap mutagenesis screen that would preclude its elimination in viable cells[10,14]. In the case of the *VPS35* and *TLE3* cell lines, we observed changes in the splice variants harboring the CRISPR-targeted exons although no detectable novel proteins emerged (Fig. 1c).

In contrast to the *TOP1* clones, the *CTNNB1* and *LRP6* cell lines exhibited no detectable change in mRNA splicing associated with the targeted exons suggesting the novel proteins are a consequence of alternative translation initiation (ATI) events presumably induced by the introduced INDELs (Fig. 1c). Consistent with this hypothesis, the mutant LRP6 protein is not glycosylated perhaps as a consequence of default expression in the cytoplasm in the absence of its N-terminal signal sequence (Supplementary Fig. 1A, C). Similarly, the novel β-catenin protein co-migrates on SDS-PAGE with an engineered β-catenin protein initiating from Met88 (Supplementary Fig. 1B). Similar events have previously been reported in transcripts with PTCs introduced proximal to the native initiation site in cancerous cells[15]. In summary, in ~50% of CRISPR-edited cell lines acquired from a commercial source, we observed unexpected changes in protein expression or mRNA splicing that challenge the notion that these reagents could be used to report the cellular effects of complete genetic ablation (Fig. 2c). Although not investigated here, conceivably the mutant proteins could also contribute to neomorphic cellular phenotypes.

**ATI and pseudo-mRNAs confound CRISPR-based gene knockout.** We had complemented our efforts to generate cells genetically null for various genes-of-interest with de novo CRISPR-Cas9-based gene targeting projects. As part of our focus on the tumor suppressor kinase LKB1, we observed the emergence of unexpected protein products—both smaller and larger proteins than the canonical protein—that were not readily explained by the presence of CRISPR-introduced INDELs (Fig. 3a–c). Given the INDELs created in *LKB1* are localized to the first protein coding exon (Fig. 3d) and the antibody recognizing the C- but not the N-terminus epitope reported the shortened LKB1 protein on SDS-PAGE (Fig. 3b, c), we concluded that an ATI event induced by CRISPR-Cas9-introduced INDELs likely resulted in an LKB1 protein lacking a portion of its N-terminal sequence (ATI LKB1 protein).

We also noted in MIA, but not HAP1 cells, a slower migrating protein recognized by LKB1 antibodies emerged in CRISPR-Cas9-edited clones with frameshift-inducing INDELs (Fig. 3c; Super LKB1 protein). The appearance of Super LKB1 protein coincided with the appearance of a new mRNA splice variant that contained a 131 bp exon not included in the transcript that encodes the canonical LKB1 protein (Fig. 3e). Consistent with this exon belonging to an *LKB1* pseudo-mRNA not previously annotated in MIA cells, the addition of cycloheximide (CHX) to disrupt NMD in parental MIA cells resulted in the emergence of an *LKB1* splice variant that includes this exon (Supplementary Fig. 2A). Thus, the same INDELs that induced a frameshift in the canonical transcript now removed a PTC from an *LKB1*

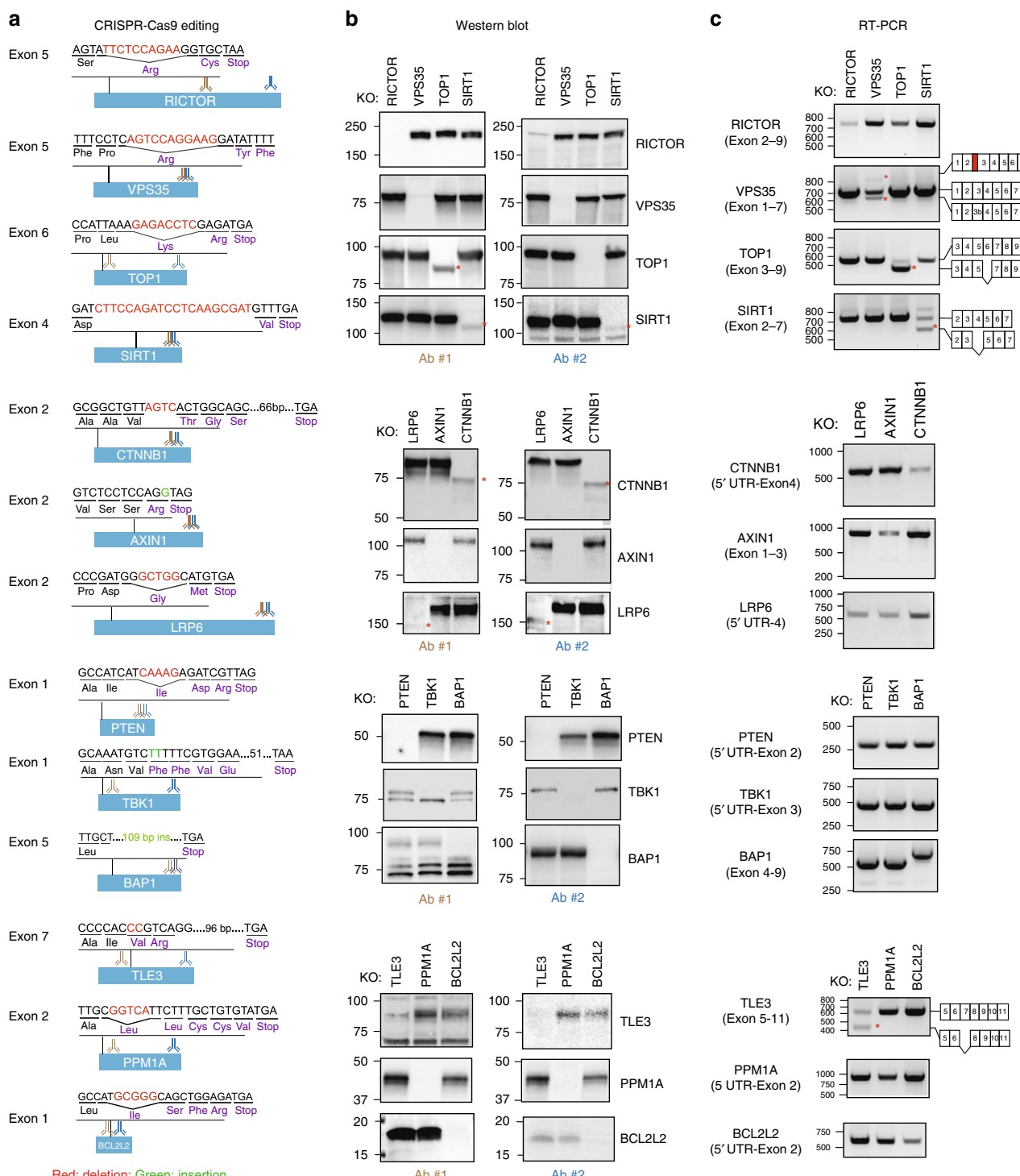

**Fig. 1** Unanticipated gene expression outcomes following on-target CRISPR editing. **a** The effect of CRISPR-introduced frameshift alterations on mRNA and protein expression was analyzed using a panel of CRISPR-Cas9-edited HAP1 cells that were commercially accessible. The targeted exon, anticipated PTC location following insertion/deletion mutation and the protein recognition sites of antibodies used in panel **b** are indicated. **b** Appearance of novel proteins in cells edited with CRISPR-Cas9. HAP1 cells were subjected to western blot analysis using two distinct antibodies. Asterisks (*) indicate novel proteins. **c** CRISPR-Cas9 gene editing induces expression of novel mRNA species. RT-PCR analysis of edited cells was performed using primers recognizing flanking exons and the amplicons generated were sequenced. Asterisks (*) indicate novel mRNA species. Source data are provided as a Source Data file

pseudo-mRNA and capacitated it for protein production (Fig. 3e). We noted that HAP1 cells did not transcribe an mRNA containing this exon thus our introduction of INDELs into exon 1 did not result in the production of the Super LKB1 protein (Supplementary Fig. 2B). An understanding of both the

transcriptome and the pseudo-transcriptome in cells is thus critical to anticipating the net effect of frameshift-inducing INDELs introduced by CRISPR-Cas9 (ref. [9]).

To understand how CRISPR-Cas9-introduced INDELs may have produced the ATI LKB1 protein, we generated cDNAs

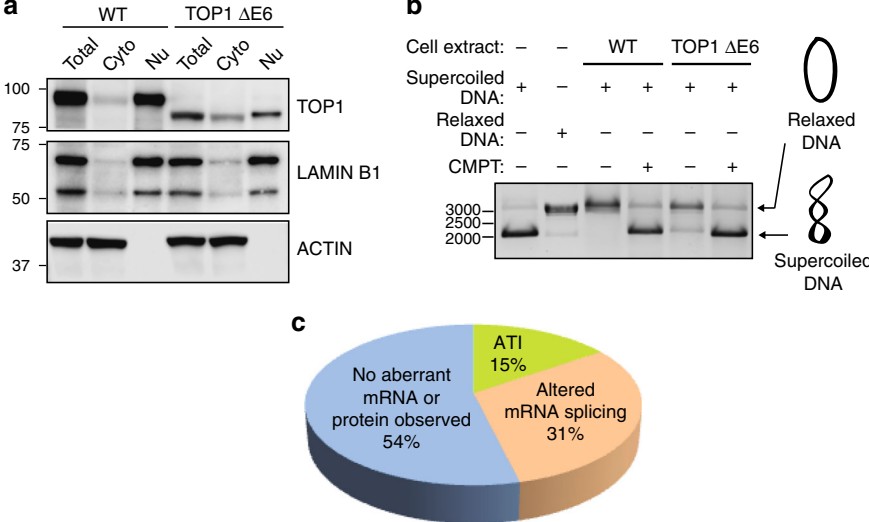

**Fig. 2** A *TOP1* gene harboring a frameshift-inducing deletion retains catalytic activity. **a** Exclusion of exon 6 (a symmetric exon) produces an internally truncated TOP1 protein (TOP1 ΔE6) with altered subcellular distribution. **b** The TOP1 ΔE6 protein can induce relaxation of supercoiled DNA. Camptothecin (TOP1 inhibitor) prevents DNA relaxation. **c** Summary of novel mRNAs or proteins observed in 13 CRISPR-edited commercial HAP1 cell lines (Horizon Discovery). Source data are provided as a Source Data file

harboring each of the two INDELs that were found in the edited cells expressing these proteins (MIA cells, clone M2) in order to remove any potential contribution of altered mRNA splicing to the production of the mutant proteins (Fig. 3f). When either INDEL was introduced into the canonical *LKB1* cDNA sequence, we observed the expression of a protein that co-migrated with the ATI LKB1 protein. This unexpected protein product also co-migrated with an engineered protein that initiates at methionine 51 (Fig. 3f). We noted that a cDNA harboring the 1 bp insertion that provoked the ATI LKB1 protein likely did not induce leaky scanning (Supplementary Fig. 3), or translational re-initiation[16] given the PTC is located downstream (3′) of the predicted ATI site. We also considered whether or not alternative secondary structures of the mutant mRNAs might induce this ATI at methionine 51 using an algorithm for modeling conserved RNA structures (Supplementary Figs. 4–6). At least using this approach, we anticipate changes in RNA folding that may influence the location of ribosomal initiation. At the same time, we also evaluated the effects of these mutations on cDNAs that encode the predicted pseudo-mRNA sequence (with the 131 bp additional exon). As anticipated, we observed the emergence of proteins that co-migrated with Super LKB1 protein given that either CRISPR-introduced mutation in a transcript with the additional 131 bp sequence would eradicate the naturally occurring PTC present in the pseudo-mRNA sequence (Fig. 3f).

**ATI suppresses NMD**. Despite the introduction of a frameshift-promoting INDEL in *LKB1*, we presumed that an ATI event, which restores codon usage to its native phase, would fail to elicit NMD during the pioneer round of translation. At the same time, having avoided destruction, the mutant mRNA is now able to support repeated rounds of translation including presumably short polypeptides initiating at the canonical start site and ending at the PTC. Given our initial western blot analysis of the LKB1 CRISPR-edited clones did not capture low molecular proteins (Fig. 3b, c), re-examination of LKB1 proteins in our CRISPR-edited clones indeed revealed the presence of a small LKB1 polypeptide. This protein (short LKB1) co-migrates with an engineered protein that initiates at the canonical start site but

terminates at the presumed PTC introduced by the INDEL (Supplementary Fig. 7A).

We compared the effects on mRNA stability of an INDEL associated with ATI with an INDEL that yielded no detectable LKB1 polypeptides (Supplementary Fig. 7B C) in order to determine if ATI suppressed NMD as a potential mechanism for promoting C-terminally truncated proteins. Comparing the levels of the two LKB1 mRNAs, we observed greater loss of the mRNA in the CRISPR-edited clone lacking any detectable ATI events (Supplementary Fig. 7D). We observed little difference induced by CHX exposure in LKB1 mRNA abundance in the ATI-associated cells when compared to parental cells suggesting that NMD is not acting on the mRNA with an ATI-provoking mutation (Supplementary Fig. 7D). On the other hand, in the case of the CRISPR-edited cell line that expresses no LKB1 polypeptides, we observed a 10-fold change in LKB1 mRNA in the presence of CHX suggesting the mutant mRNA in this case is subject to robust NMD action (Supplementary Fig. 7D). In total, we observed the production of three polypeptides in lieu of the canonical LKB1 protein following the introduction of a frameshift-inducing INDEL: Super LKB1, ATI LKB1, and Short LKB1 (Supplementary Fig. 7E). More generally, our observations also suggest that introducing INDELs early in the transcript increases the potential for an ATI event that is able to clear off all of the splice junction complexes during the pioneer round thus enabling the synthesis of polypeptides with truncations in the C-terminal sequence.

**Exon symmetry influences CRISPR outcomes**. In the analysis of our assembled HAP1 cell line panel, we also observed ~30% of the clones exhibited exclusion of the targeted exon in the mRNA. Exons are replete with splicing regulatory motifs including exon splicing enhancers and suppressors (ESEs and ESSs, respectively). These degenerate hexameric sequences dictate the extent to which exons are included within a transcript[12,17]. We suspected that exon exclusion was at least in part due to the disruption of ESEs by an INDEL event. As part of our efforts focused on studying the SUFU tumor suppressor protein, we had generated a collection of cells that presumably were null for *SUFU* based on western blot analysis (Fig. 4a, b). Yet, we noted that many of these clones exhibited exclusion of the targeted exon (Fig. 4c). The extent of

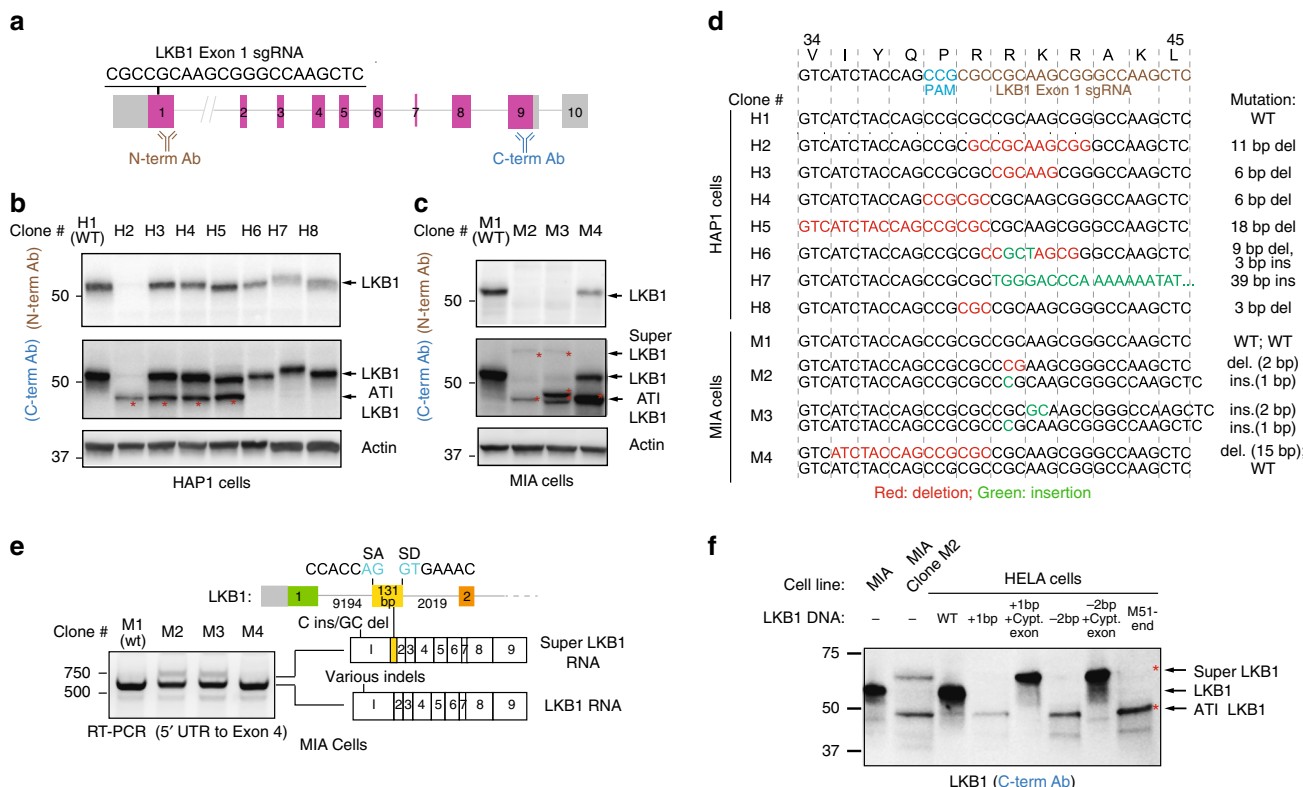

**Fig. 3** ATI and pseudo-mRNAs contribute to foreign protein production in CRISPR-edited cell line. **a** Genomic structure of the *LKB1* gene and the exonic sequence targeted by the *LKB1* exon 1 sgRNA. **b** Emergence of a small LKB1 protein (ATI LKB1) as a consequence of CRISPR-Cas9 gene editing. Lysates generated from CRISPR-edited HAP1 clones were subjected to western blot analysis using two distinct LBK1 antibodies recognizing either N- and C-terminus localized epitopes. **c** Western blot analysis of CRISPR-Cas9-edited MIA clones reveals the appearance of a large LKB1 protein (Super LKB1) in addition to the ATI LKB1 protein. **d** Genomic sequences of CRISPR-Cas9-edited HAP1 and MIA clones reveal on-target insertion/deletion mutations in the *LKB1* gene. Predicted gene alteration for each clone is indicated. **e** CRISPR-Cas9-introduced INDELs are associated with the expression of an LKB1 pseudo-mRNA transcript. RT-PCR analysis was performed using primers mapping to 5′ UTR and exon 4 in LKB1 to generate amplicons from the cDNA of CRISPR-Cas9-edited clones. MIA clones M2 and M3, which express Super LKB1 protein, harbor an mRNA species that includes an additional exon. The 131 bp additional exon contains canonical splice acceptor and donor sequences. **f** A cDNA expression strategy for understanding allele-specific CRISPR-introduced INDELs on protein expression provides evidence for ATI. *LKB1* and Super *LKB1* cDNA expression constructs harboring genomic alterations found in *LKB1* of MIA Clone M2 were introduced into HELA cells that lack endogenous *LKB1* expression. The 1 bp insertion or 2 bp deletion in the Super *LKB1* cDNA result in proteins that co-migrate with the Super LKB1 protein observed in MIA Clone M2. On the other hand, the same mutations in *LKB1* cDNA give rise to proteins that co-migrate with the ATI LKB1 protein found in Clone M2, and with the protein that initiates at Met51. Source data are provided as a Source Data file

exon exclusion notably differs suggesting other factors, perhaps RNA structure changes that contribute to exon splicing regulation, also may be compromised by the introduction of an INDEL at this position within the SUFU mRNA. We identified a cluster of potential ESEs in the targeted *SUFU* exon that was likely impacted by the INDEL in these clones (Fig. 4d). No ESSs were identified in this case. To determine how reliably we can induce exon exclusion by impacting a predicted ESE, we introduced INDELs at putative ESEs found in other *SUFU* exons and performed similar analysis of the protein and mRNA in RMS13 cell line (Fig. 4e–l). In every instance, we observed exon exclusion by targeted disruption of a putative ESE.

When all the clones presented so far from both commercial and de novo engineered were considered with respect to predicted impact on an ESE and exon exclusion, we observed a strong correlation between these two events (Fig. 4m; Supplementary Fig. 8). A subset of the clones exhibiting alternative splicing also expressed novel polypeptides (see *TOP1* and *SIRT1*; Fig. 1b). We noted in both these cases that the exons were symmetric—meaning the exon harbors a nucleotide number in multiples of three, and exclusion of this exon would result in a transcript that

retains the original reading frame. In the case of the *SUFU* clones, the majority of exons skipped were asymmetric thus likely resulting in the lack of protein expression. However, we noted one targeted and skipped exon (exon 2) was symmetric yet the resulting transcript failed to generate a detectable protein perhaps due to misfolding of the mutant protein (Fig. 4e, f). Indeed, the skipped exon encodes part of an intrinsically disordered region of the protein that is essential for interaction with members of the pro-survival BCL2 family members[18]. From these *SUFU* clones, we expect that decreased SUFU mRNA seen in CRISPR-edited cells was due to NMD provoked by the introduction of a frameshift-inducing INDEL, or exclusion of the targeted asymmetric exon and the introduction of a PTC in an NMD-enabling position within the gene.

**CRISPinatoR.** Purposeful disruption of ESEs in asymmetric exons could improve gene knockout efficiency given that even INDELs that fail to alter the coding frame would have a second opportunity for introducing a PTC by skipping the exon altogether. In addition to the evidence provided here, the ability of mutations in ESEs to alter mRNA splicing have been documented

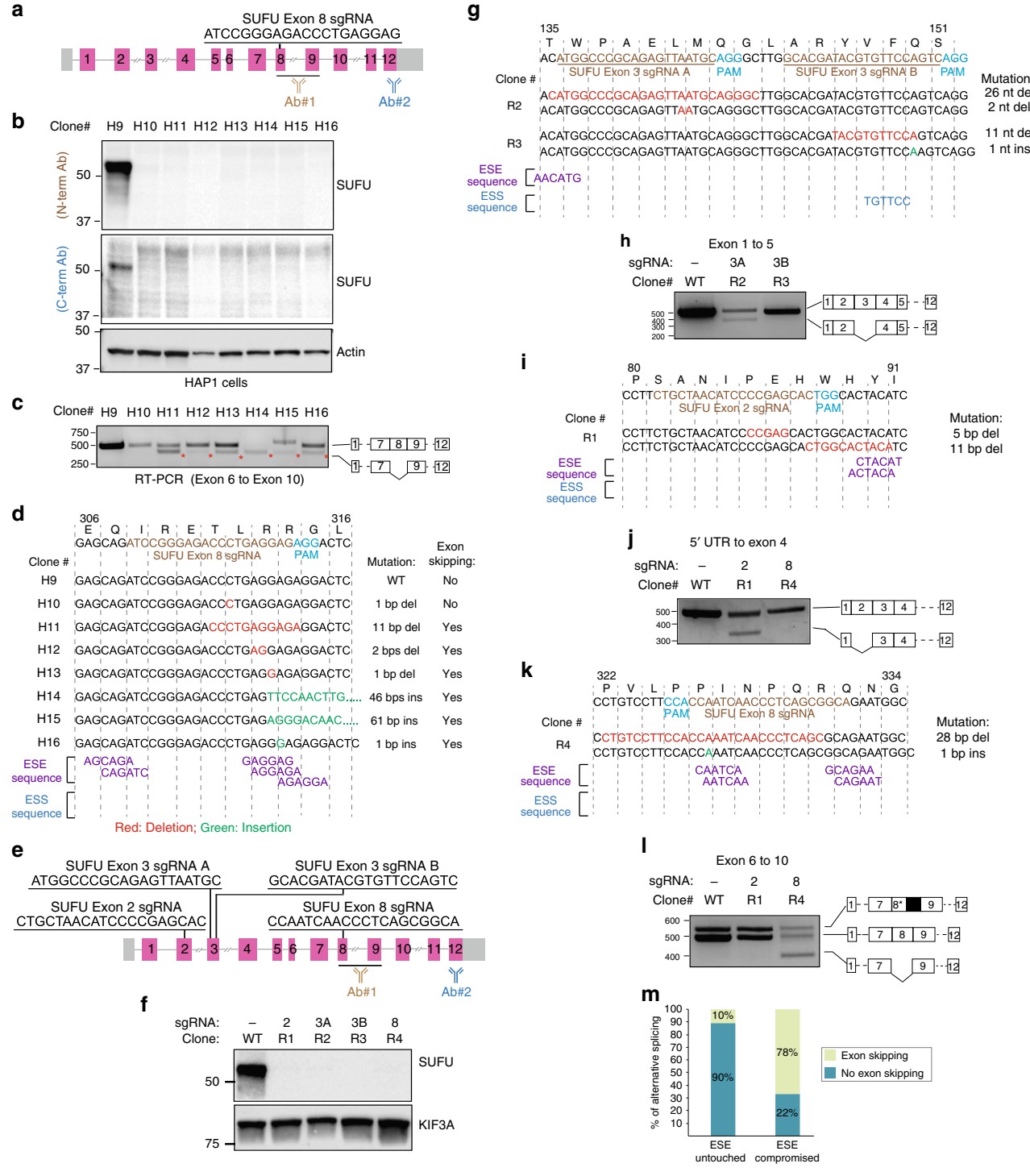

elsewhere[19,20]. To systematize this strategy, we developed the CRISPinatoR, a website that identifies asymmetric exons found in a given gene and CRISPR-Cas9 guide sequences that help to deliver double-stranded breaks within proximity of a putative ESE (Fig. 5a, Supplementary Fig. 9). At the same time, the portal could be used to induce the skipping of an exon harboring a deleterious mutation in order to generate a novel protein that may retain function. We note that when analyzing genome-wide CRISPR libraries, that ratio of guides targeting symmetric and asymmetric exons was fairly consistent, suggesting that these algorithms do not factor in potential gene elimination efficiency based on exon symmetry (Supplementary Fig. 10A, B). Similarly, the

CRISPinatoR could be used to re-evaluate previously reported phenotypes using CRISPR-Cas9 based on the potential for the sgRNA for inducing exon skipping.

**Targeting RNA-regulatory elements for gene knockout agendas.** We tested the ability of CRISPinatoR to design guides that induce exon skipping for either degradation of mRNA or production of novel protein-encoding mRNAs by targeting asymmetric or symmetric exons, respectively. Using the WNT receptor LRP5 as a case study, we asked the CRISPinatoR to identify sgRNAs that presumably would be able to induce exon skipping

**Fig. 4** Compromised ESEs account for INDEL-induced exon skipping. **a** Genomic structure of *SUFU* and exonic sequence targeted by *SUFU* exon 8 sgRNA. The recognition sites of antibodies used in panel **b** are indicated. **b** Western blot analysis of HAP1 cells edited with *SUFU* exon 8 sgRNA shows no detectable expression of SUFU. **c** Exon skipping is prevalent in CRISPR-Cas9-edited SUFU clones. RT-PCR analysis using primers flanking exons 6 and 10 of *SUFU* in CRISPR-Cas9-edited SUFU clones. Sequencing of amplicons reveals exon skipping in all of clones except clones H9 and H10. **d** Disruption of exon splicing enhancers (ESEs) by CRISPR-introduced INDELs triggers skipping of the edited exons. Genetic mutation and the presence/absence of exon skipping events for each clone are indicated. Putative ESEs were identified using the RESCUE-ESE web server. **e** Multiple sgRNA sequences located in symmetric or asymmetric exons of the *SUFU* gene used for targeted disruption of ESEs. **f** sgRNAs described in "**e**" were used to edit the *SUFU* gene in RMS13 cells. Western blot analysis of lysates derived from the CRISPR-Cas9-edited RMS13 clones show no detectable SUFU protein. **g** Genomic sequences of RMS13 clones edited with *SUFU* exon 3 sgRNAs. CRISPR-introduced mutations and putative exon splicing enhancer (ESE) and exon splicing silencer (ESS) sequences are indicated. **h** RT-PCR analysis and cDNA sequencing result of clones R2 and R3 using primers flanking exon 1 and 5. **i** Genomic sequences of RMS13 clones edited with *SUFU* exon 2 sgRNA. CRISPR-introduced mutations and putative ESE/ESS sequences are indicated. **j** RT-PCR analysis and cDNA sequencing result of clones R1 and R4 using primers flanking 5′ UTR and exon 4. **k** Genomic sequences of RMS13 clones edited with *SUFU* exon 8 sgRNA. CRISPR-introduced mutations and putative ESE and ESS sequences are indicated. **l** RT-PCR analysis and cDNA sequencing result of the clones R1 and R4 using primers flanking exon 6 and exon 10. **m** Disruption of ESE code is highly reliable in anticipating CRISPR-Cas9-induced exon skipping. Twenty-four CRISPR-Cas9-edited cell lines with different mutations were analyzed for the presence/absence of exon skipping events and changes in ESE sequences due to CRISPR-introduced INDELs. Source data are provided as a Source Data file

in each exon class (Fig. 5b). We identified clones that harbored INDELs at the anticipated LRP5 exonic sequence by targeted sequencing of isolated genomic DNA (Fig. 5c). Using RT-PCR analysis coupled with targeted sequencing, we observed exon skipping in clones associated with both guides (Fig. 5d; Supplementary Fig. 11). We observed an absence of LRP5 protein in the clone exhibiting exclusion of an asymmetric exon (Fig. 5e). However, in the clone exhibiting exclusion of a symmetric exon, we observed the appearance of a faster migrating protein (Fig. 5e). We confirmed that this new protein retains glycosyl moieties, suggesting that its signal sequence localized to the N-terminus is intact unlike in the case of the *LRP6* edited HAP1 clone (Fig. 5f; Supplementary Fig. 1A, C). The presence of a secreted protein and evidence for skipping of the CRISPR-targeted exon suggest that the novel LRP5 protein formed would harbor a compromised β-propeller domain—one of two that contributes to WNT3A binding (Fig. 5g). Indeed, we observed response of a clone expressing the truncated LRP5 protein to exogenously supplied WNT conditioned medium using a WNT pathway reporter (Fig. 5h). The weakened response compared to WT HAP1 cells likely reflects reduced total LRP5 protein levels and/or reduced WNT-binding affinity with deletion of exon 16 sequence. On the other hand, the cell expressing the LRP5 mRNA excluding the CRISPR-edited asymmetric exon showed a loss of WNT pathway response consistent with the absence of LRP5 protein production from an mRNA lacking an asymmetric exon (Fig. 5h).

## Discussion

The microRNA-like behavior of short interference RNAs (siR-NAs) has long posed a challenge to using RNA interference (RNAi) for selective gene product ablation in both early discovery and therapeutic settings[21,22]. Whereas this issue is not inherent in DNA-editing systems such as CRISPR-Cas9, we show here that this technological advantage is offset by the unanticipated effects stemming from the on-target changes that impact the regulation of the RNA product and the translation of the protein it encodes. Exon skipping events, for example, associated with CRISPR have been previously observed although the mechanistic basis for these phenomena was not well-understood[23]. Our incomplete understanding of RNA splicing regulatory mechanisms, inability to accurately predict RNA structural changes introduced by INDELs, and limited accounting of the pseudo-transcriptome challenge our ability to anticipate transcriptional and translation outcomes as a consequence of introducing INDELs in exonic sequences (Fig. 6). Indeed, we assume these trials extend to other INDEL-producing gene editing systems that have been applied in human cells such as the CRISPR endonuclease Cpf1 (ref. [24]).

A number of considerations in guide design could be installed in our design workflow to increase the fidelity of DNA sequencing information for predicting protein translational outcomes. A map of RNA-regulatory motifs (such as ESEs) that might be impacted by a CRISPR-Cas9-delivered INDEL such as that generated by the CRISPinatoR for the human genome could help in improving gene elimination or protein engineering campaigns. We acknowledge that the impact of RNA structure and possibly other determinants that can influence the function of regulatory sequences involved in RNA splicing, for example, are not accounted for by our database. At the same time, an understanding of lineage-associated pseudo-transcripts that would be edited alongside the intended target transcripts would also help to anticipate the emergence of novel protein products such as Super LKB1 from conversion of a pseudo-mRNA to a protein-encoding mRNA.

Perhaps the most daunting challenge that we encountered from our analysis of CRISPR-edited cell lines is the emergence of IRESs likely due to INDEL-induced changes in RNA structure. We anticipate that the number of ATI events associated with INDELs will be higher than what is reported here given the shortage of antibodies useful for detecting native as well as potentially truncated proteins that emerge from ATI. In this regard, the use of translation inhibitors such as CHX combined with RT-PCR could be a simple method to flag mRNAs that harbor CRISPR-Cas9-introduced frameshift-inducing INDELs yet for reasons including ATI subversion are not substrates for NMD. Although we have attempted to account for the ATI events we observed in our LKB1 gene editing projects using an in silico RNA structure prediction strategy, admittedly other factors such as potential changes in RNA-binding protein interactions could contribute to alterations in translation initiation sites.

Our observations also have implications for the use of INDEL-based genome editing tools for gene rescue efforts where induced exon skipping can excise sequences that harbor a mutation thus producing a viable gene[25]. These outcomes are currently achieved by using two CRISPR guides that flank a mutated exon[26–28], or target an exon-specific splice junction using a single guide[29]. However, the ability to use a single guide targeting an ESE to achieve a similar outcome should reduce the dangers of using two CRISPR guides and expand the number of single guide options with acceptable off-target risks. In this regard, guides identified by the CRISPinatoR targeting ESEs found in symmetric exons could be used to systematically identify such opportunities in genes involved in disease. Needless to say, mRNA splicing is a complex phenomenon and this approach should serve as a starting point

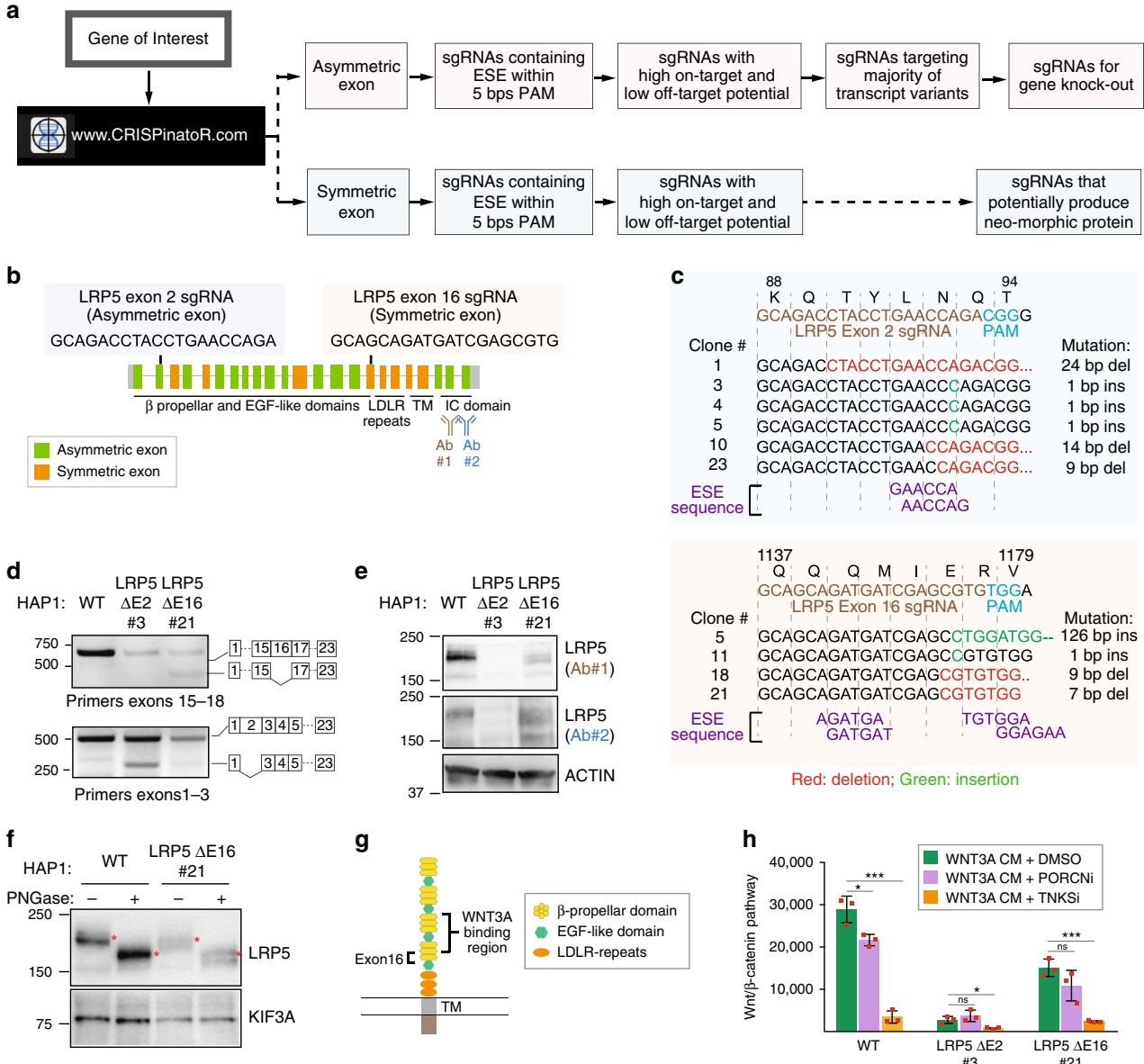

**Fig. 5** Targeting RNA-regulatory elements for gene knockout agendas. **a** CRISPinatoR: a web-based guide RNA design tool that utilizes targeted ESE disruption for achieving gene elimination. CRISPinatoR identifies sgRNA sequences that target ESEs in asymmetric exons and calculates off-targeting potential and the number of splice variants impacted by the sgRNAs. A scoring system that integrates all three parameters is used to provide sgRNAs with high gene knockout potential. **b** Genome structure of the *LRP5* gene and sgRNA sequences targeting the asymmetric exon 2 and the symmetric exon 16. **c** Genomic sequencing results of HAP1 clones edited using *LRP5* exon 2 and exon 16 sgRNAs. CRISPR-introduced mutations and the putative ESE sequences are indicated. **d** Exclusion of an asymmetric or a symmetric exon with INDEL-induced changes to the putative ESE sequences. RT-PCR analysis and cDNA sequencing result of HAP1 cells edited with *LRP5* exon 2 and exon 16 sgRNAs. **e** Targeted ESE disruption in asymmetric exon increases gene knockout potential. Western blot analysis of HAP1 clones edited with *LRP5* exon 2 sgRNA (Clone 21) and exon 16 sgRNA (Clone 3) was probed with two distinct antibodies indicated in "**b**". ESE disruption in symmetric exon 2 produces internally truncated in-frame LRP5 protein. **f** The internally truncated LRP5 protein is glycosylated. Lysates derived from WT or LRP5 ΔE16 HAP1 cells were incubated with the deglycosidase PNGase F then subjected to western blot analysis. **g** Exclusion of *LRP5* exon16 would delete a sequence adjacent to the WNT3A binding domains. **h** The LRP5 ΔE16 protein formed post skipping of a symmetric exon is functionally active. WNT/β-catenin pathway activity in response to WNT3A conditioned medium (WNT3A CM) was measured for HAP1 WT, LRP5 ΔE2, and LRP5 ΔE16 cells. WNT pathway inhibitors WNT974 (PORCNi) and IWR1 (TNKSi) serve as negative and positive control, respectively. All error bars represent mean of triplicates ± s.d. The experiment was repeated three times with similar results. Statistical testing was performed using Student's *t*-test, *P < 0.05, **P < 0.01, ***P < 0.001, ****P < 0.0001. Source data are provided as a Source Data file

for evaluating the potential effects of INDELs on mRNA regulation.

## Materials and methods

**Cell lines and reagents**. WT and CRISPR-edited HAP1 knockout commercial cell lines were purchased from Horizon Discovery (Supplemental Table 1). HELA, MIA PaCa-2, and RMS13 cell lines were purchased from ATCC. Hela cells (listed in the database of commonly misidentified cell lines, ICLAC) lack endogenous *LKB1* expression and therefore was used in an experiment to monitor the protein expression encoded LKB1 cDNA constructs harboring CRISPR-introduced INDELs. All the cell lines were tested for mycoplasma contamination. None of the cell lines were authenticated since all the cell lines were directly purchased from commercial suppliers.

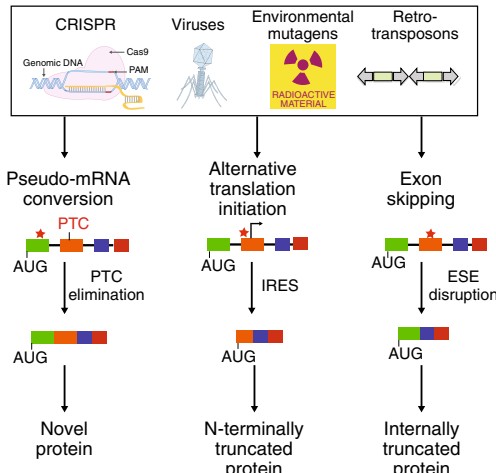

**Fig. 6** Cellular mechanisms for countering INDEL effects revealed by CRISPR failures. INDELs introduced by Cas9 and other enzymes used for gene editing elicit transcriptional and translational responses that may have evolved to buffer the transcriptome and proteome against common environmental insults

Puromycin was purchased from Fisher Scientific (ICN10055225). Cycloheximide was purchased from EMD Millipore (239765). NE-PER Nuclear and Cytoplasmic extraction reagent (78833) was purchased from ThermoFisher.

**Western blot analysis**. Cell lysates were generated with PBS/1% NP40 buffer supplemented with protease inhibitor cocktail (Sigma Cat. No. S8820). Protein sample loading buffer was added to cell lysates and proteins were separated on SDS-PAGE (BioRad Criterion TGX Precast Gel). The following primary antibodies were used for immunoblotting at the indicated dilutions: Cell Signaling Technology: AXIN1 (2087 and 2074; 1:1000), BAP1 (13271; 1:1000), BCL2L2 (2724; 1:1000), LRP6 (2560; 1:1000), PTEN (9552; 1:1000), RICTOR (9476; 1:1000), SIRT1 (2493; 1:1000), TBK1 (3504; 1:1000), TLE3 (4681; 1:1000), LKB1 (3050; 1:1000), SUFU (2520; 1:1000), LRP5 (5731; 1:1000). Bethyl Laboratories: BAP1 (A302-243A-T; 1:1000), RICTOR (A300-459A; 1:1000), SIRT1 (A300-688A; 1:1000), TOP1 (A302-589A and 302-590A; 1:1000), VPS35 (A304-727A; 1:1000). Abcam: TBK1 (40676; 1:1000), PPM1A (14824; 1:1000), SUFU (52913; 1:1000). Santa Cruz Biotechnologies: LRP6 (sc-25317; 1:1000), TLE3 (sc-9124; 1:1000), LKB1 (sc-374334; 1:1000), LRP5 (sc-390267; 1:1000). LS BioScience: BCL2L2 (LS-C382259-100; 1:1000), PPM1A (LS-C169090-100; 1:1000). Sigma-Aldrich: CTNNB1 (C2206; 1:1000). Invitrogen: PTEN (44-1064; 1:1000). BD Biosciences: CTNNB1 (610153; 1:1000). Genetex: VPS35 (GTC108058; 1:1000).

**Transfection of sgRNAs**. $1 \times 10^6$ MIA PaCa-2 or HAP1 cells were seeded per six-well plate and co-transfected with 0.5 μg pCas-Guide plasmid using Effectene transfection reagent (Qiagen). Twenty-four hours after transfection, cells were trypsinized and plated in 150 mm² culture dishes in various dilutions for clonal selection.

**Clonal isolation of CRISPR-edited cells**. Cells in 150 mm² culture plates were treated with 0.5 μg/ml of puromycin in order to enrich for cells expressing Cas9. Puromycin selection was maintained for 10 days after which single colonies were isolated and grown in a 96-well plate. Cells from single colonies were passaged multiple times until sufficient cells were available for analyzing genomic DNA, RNA, and protein.

**Genomic DNA extraction and genomic sequencing**. Genomic DNA was extracted from the CRISPR-edited cells using Genomic DNA Minikit (Bioland Scientific) according to the manufacturer's instructions and used as a template for PCR amplification. PCR primers encompassing the CRISPR-targeted region were designed. PCR was performed with GoTaq Green Master Mix (Promega M7122) with following conditions: 98 °C for 2 min (initial denaturation), 25 cycles of 98 °C for 30 s, 56 °C for 30 s, 72 °C for 30 s (denaturation, annealing, extension), and final 70 °C for 5 min (final extension). Gel electrophoresis in a 1.5% agarose gel was performed and the PCR products were purified from the gel using QIA Quick PCR Purification Kit (Qiagen) and cloned into pCR-TOPO plasmid using TOPO TA cloning kit for Subcloning (ThermoFisher Scientific). pCR-TOPO plasmids containing genomic DNA sequences were transformed into TOP10 competent cells and individual colonies were selected and sequenced.

**RNA extraction and analysis**. RNA extraction was performed using RNeasy Mini Kit (Qiagen) according to the manufacturer's instructions. cDNA synthesis was performed on 1 μg of RNA using the ProtoScript First Strand cDNA Synthesis Kit (Promega). Primers recognizing exons flanking the CRISPR-targeted exon (Supplemental Table 3) were used to amplify the cDNA sequences isolated from the CRISPR-edited cells. PCR products were electrophoresed in a 1% agarose gel and the gel bands were isolated using QIA Quick PCR Purification Kit (Qiagen). Isolated DNA was cloned into pCR-TOPO plasmids using the TOPO TA cloning kit (ThermoFisher Scientific), and clones were sequenced at the UTSW Sequencing Core.

**RNA secondary structure modeling**. Conserved secondary structures were modeled using TurboFold II[30]. The full-length sequences of the five clones without ATI (HAP1 clones H1, H6, H7, H8 and 3 bp substitution) and eight clones with ATI (HAP1 clones H2, H3, H4, H5 and MIA clones M2.1, M2.2, M3.1, M4.1) were modeled separately. Default parameters were used with TurboFold II. The resulting secondary structures of each CRISPR clone were mapped to the manual alignment of all clones in dot-bracket format.

**Nuclear and cytoplasmic fractionation**. WT and TOP1 ΔE6 cells were washed with PBS. Nuclear and cytoplasmic extract were prepared using NE-PER Nuclear and Cytoplasmic Extraction Reagents according to the manufacturer's protocol (ThermoFisher).

**ESE selection and design of CRISPinatoR**. ESE sequences were collected from Ke et al.[31] (top 200 hexamers based on ESEseq score) and Rescue-ESE[32] (238 hexamers). Forty-three sequence motifs for RNA-binding proteins associated with splice regulation were also included[8]. Redundant hexamers and motifs were removed and finally 440 motifs were used to generate a reference collection of ESE sequences. We scored the frequency of ESE sequences that were potentially impacted by guides found in commonly used CRISPR libraries (GeckoV2, Avana, TKO V3 and Sanger). The target exon for each sgRNA and the symmetry of the target exon was identified using genomic annotation from Ensembl[33]. The number of ESEs within a given 20 bp sgRNA sequence was annotated. To score the off-target potential for each sgRNA candidate, we modified the bwa source code[34]. The total 23 bp sequence (20 bp sgRNA + 3 bp PAM) was aligned to the hg19 reference genome allowing up to three mismatches. The variable bp in the 5′ most position of the PAM sequence was not considered for mismatch scoring. An off-target score (0–100) for the sgRNA was calculated by a method used in Hsu et al.[35] and a cutoff of 80 was considered acceptable.

**Reporting summary**. Further information on research design is available in the Nature Research Reporting Summary linked to this article.

## Data availability
The data supporting the findings of this study are available from the corresponding author upon request. The source data containing images of uncropped blots used in this paper are provided as a Source Data file.

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

## Acknowledgements

This work was supported in part by Welch Foundation (I-1665 to L.L.), CPRIT (RP130212 to L.L.), the National Cancer Institute (1R01 CA168761 to J.K.), the American Cancer Society (RSG-16-090-01-TBG to J.K.), and the National Institutes of Health (R01GM076485 to D.H.M.).

## Author contributions

L.L. and T.H.H. supervised the study. R.T., J.T.P., X.W., Q.B., Z.T. and J.H. performed the experiments. Y.Y., J.R.C. and T.H.H. executed computational studies and wrote the algorithm for the CRISPinatoR. D.H.M. and J.K. provided help with data interpretation. R.T., J.K. and L.L. wrote the manuscript.

## Additional information

**Competing interests:** The authors L.L., R.T., T.H.H., Y.Y., J.T.P. and Q.B. are named inventors on a patent (under consideration: 16/003683) applied for by University of Texas Southwestern Medical Center, Dallas, on behalf of the inventors that covers strategies for inducing or avoiding exon skipping using CRISPR — specifically the CRISPinator algorithm and targeting exon splicing enhancers using INDELs to induce exon skipping with a single CRISPR guide strategy. The remaining authors declare no competing interest.

