## [Peer Review File · Nature Communications]

Reviewers' comments:

Reviewer #1 (Remarks to the Author):

The authors present some interesting - if rather anecdotal - examples of how CRISP/Cas9 gene editing can result in the production of new RNA isoforms due to indel-induced changes in splicing or translation start sites. I found the examples interesting and quite thought-provoking and it is important that researchers are aware that gene editing can have unanticipated outcomes including the production of non null or (potentially) neomorphs.

What I found less convincing about the study was:

(i) the investigation of the mechanisms underlying the transcript isoform changes. In particular, the 'mechanism' for the change in translation initiation start (IRES) is speculative based on the (notoriously unreliable) computational prediction of RNA secondary structures. Without additional experimental tests I think this section should be removed from the results section and presented in the discussion section as a hypothesis that needs to be tested.

(ii) the CRISPinatoR tool for targeting guide sequences to asymmetric exons and putative exonic splicing enhancers. Asymmetric exons can obviously be identified from sequence, but what is the evidence that the ESE prediction helps to identify guides that will disrupt splicing? If there is one thing that deep mutagenesis has taught us it is that the current computational methods are very poor for predicting the effects of exonic mutations on splicing.

(iii) "Our results using CRISPR/Cas9-introduced INDELS reveal facets of an epigenetic genome buffering apparatus that likely evolved to mitigate the impact of such mutations introduced by pathogens and aberrant DNA damage repair". "A genome buffering apparatus" seems wildly speculative, vague and, from a population genetics perspective, unlikely to be correct. Certainly should not be in the abstract.

Reviewer #2 (Remarks to the Author):

The authors screened a panel of 13 commercially available HAP1 cell lines in which different individual genes had been targeted for frameshifting INDELS by CRISPR/Cas9. Using two independent antibodies for each gene, the authors showed that for 4 of 13 lines, shorter proteins were detected. RT-PCR demonstrated exon skipping of the targeted exon as previously described (see Genome Biol 18:109). For other targeted genes, pre-mRNA splicing was not affected but protein was expressed from internal translation start sites. The predicted aberrant proteins were produced from cDNAs containing the INDELS confirming their results. A role for secondary structure is proposed but not demonstrated. The SUFU gene was found to be sensitive to ESE disruption leading to skipping of different exons making the point of the frequency of required ESEs and the importance of whether an exon is or is not in frame. To this end they present a gRNA design tool to target exons the skipping of which would alter the reading frame and promote NMD; this is a useful addition. These are important findings since CRISPR-induced KO cell lines are often assumed to be null with regard to the targeted allele. The fact that such a high percentage of commercial cell lines are unexpectedly expressing proteins from the targeted allele is a significant finding. There are several issues with the manuscript, however, as outlined below.

It is not clear where the 50% come from for this statement, "In summary, in ~50% of CRISPR-edited cells acquired from a commercial source we observed unexpected changes in protein expression or

mRNA splicing..."

The point of Figure 2 that nuclear TOP1 is substantially reduced by the loss of exon 6 is not convincing for two reasons. First the western and the graph shows that TOP1 is still predominantly nuclear and there is no stats informaton for the graph. Overall the larger point is that there is any protein at all and it is likely to be catalytically active. Fig 2 adds little and in fact is a distraction in my opinion.

Pseudo-messenger RNA is a vague term that is not generally used. In fact a pubmed search comes up with only the one paper cited in this manuscript from 2006. Aberrant mRNA or aberrantly spliced mRNA would be better terms. Similarly the term exon symmetry or asymmetric exon is not standard and therefore not clear at first reading – it refers to whether an exon is a multiple of three nucleotides for which exclusion maintains the mRNA reading frame.

The analysis of the role of secondary suggests possibilities ("may facilitate ATI", "may cause stalling") but a role for secondary structure is not really demonstrated by mutations that disrupt then re-establish to determine whether there are the predicted effects.

As noted above there are issues of clarity in the text and more direct simple statements would be helpful. For example, the main point for this is not clear – their presentation of the effect of the 1 bp insertion and why is hard to figure out. "We noted that a cDNA harboring the 1 bp insertion that provoked the ATI LKB1 protein was not associated with a redistribution of the initiation site suggesting leaky scanning is not likely responsible for the ATI event (Supp. Fig. 3). Given that the PTC is located 3' to the predicted ATI site, we also assume this is not a re-initiation phenomenon in which translational termination is followed by the ribosome re-launching translation at a secondary AUG codon." Also what specifically is "pressure-tested" rather than just tested?

The test of the CRISPinatoR design tool was modest targeting only two exons. ESE and ESS prediction algorithms are imperfect and this should be described since the paper presents identification of these elements as a given.

Response to reviewers' comments:

Reviewer #1 (Remarks to the Author):

The authors present some interesting - if rather anecdotal - examples of how CRISP/Cas9 gene editing can result in the production of new RNA isoforms due to indel-induced changes in splicing or translation start sites. I found the examples interesting and quite thought-provoking and it is important that researchers are aware that gene editing can have unanticipated outcomes including the production of none null or (potentially) neomorphs.

Response: We thank the reviewer for this feedback.

What I found less convincing about the study was:

(i) the investigation of the mechanisms underling the transcript isoform changes. In particular, the 'mechanism' for the change in translation initiation start (IRES) is speculative based on the (notoriously unreliable) computational prediction of RNA secondary structures. Without additional experimental tests I think this section should be removed from the results section and presented in the discussion section as a hypothesis that needs to be tested.

Response: We have moved the computational prediction of RNA changes induced by INDELS to the supplement and limited our discussion of this exercise in the body of the text.

(ii) the CRISPinatoR tool for targeting guide sequences to asymmetric exons and putative exonic splicing enhancers. Asymmetric exons can obviously be identified from sequence, but what is the evidence that the ESE prediction helps to identify guides that will disrupt splicing? If there is one thing that deep mutagenesis has taught us it is that the current computational methods are very poor for predicting the effects of exonic mutations on splicing.

Response: While we do agree with the reviewer that the effects of mutations on mRNA splicing are difficult to anticipate, we also offer that the data supports a utility of the Crispinator algorithm for guiding CRISPR-based engineering projects. We acknowledge the reviewer's concerns and have also added some caveats in the discussion of this data and with respect to the utility of the CRISPinator (see last sentence of DISCUSSION for example).

(iii) "Our results using CRISPR/Cas9-introduced INDELS reveal facets of an epigenetic genome buffering apparatus that likely evolved to mitigate the impact of such mutations introduced by pathogens and aberrant DNA damage repair". "A genome buffering apparatus" seems wildly speculative, vague and, from a population genetics perspective, unlikely to be correct. Certainly should not be in the abstract.

Response: We have removed this line from the abstract.

Reviewer #2 (Remarks to the Author):

The authors screened a panel of 13 commercially available HAP1 cell lines in which different individual genes had been targeted for frameshifting INDELS by CRISPR/Cas9. Using two independent antibodies for each gene, the authors showed that for 4 of 13 lines, shorter proteins were detected. RT-PCR demonstrated exon skipping of the targeted exon as previously described (see Genome Biol 18:109). For other targeted genes, pre-mRNA splicing was not affected but protein was expressed from internal translation start sites. The predicted aberrant proteins were produced from cDNAs containing the INDELS confirming their results. A role for secondary structure is proposed but not demonstrated. The SUFU gene was found to be sensitive to ESE disruption leading to skipping of different exons making the point of the frequency of required ESEs and the importance of whether an exon is or is not in frame. To this end they present a gRNA design tool to target exons the skipping of which would alter the reading frame and promote NMD; this is a useful addition. These are important findings since CRISPR-induced KO cell lines are often assumed to be null with regard to the targeted allele. The fact that such a high percentage of commercial cell lines are unexpectedly expressing proteins from the targeted allele is a significant finding. There are several issues with the manuscript, however, as outlined below.

Response: We thank the reviewer for this feedback.

It is not clear where the 50% come from for this statement, “In summary, in ~50% of CRISPR-edited cells acquired from a commercial source we observed unexpected changes in protein expression or mRNA splicing...”

Response: In Fig. 2D we summarize the findings from analyzing the 13 purchased cell lines using Western blotting and RT-PCR that support our statement found in the abstract and elsewhere in the manuscript. The basis for our statement regarding the number of unexpected changes to mRNA, protein, or both in these lines can be found there.

The point of Figure 2 that nuclear TOP1 is substantially reduced by the loss of exon 6 is not convincing for two reasons. First the western and the graph shows that TOP1 is still predominantly nuclear and there is no stats information for the graph. Overall the larger point is that there is any protein at all and it is likely to be catalytically active. Fig 2 adds little and in fact is a distraction in my opinion.

Response: We have refocused the discussion regarding the TOP1 “KO” cell lines on TOP1 activity rather than subcellular localization. We have kept Fig. 2A untouched but only for the purpose of confirming the expression of a truncated TOP1 protein in the “KO” cell line.

Pseudo-messenger RNA is a vague term that is not generally used. In fact a pubmed search comes up with only the one paper cited in this manuscript from 2006. Aberrant mRNA or aberrantly spliced mRNA would be better terms. Similarly the term exon symmetry or asymmetric exon is not standard and therefore not clear at first reading – it refers to whether an exon is a multiple of three nucleotides for which exclusion maintains the mRNA reading frame.

Response: We thank the reviewer for this feedback. However, “aberrantly spliced mRNA” also suggests these mRNAs have no cellular function except that they fail to produce a canonical protein. As recently described by Ma et al Nature 2019 April 568 (7751: 259-263) and El-Brolosy et al Nature 2019 April 568 (7751): 193-197, RNAs subjected to non-sense mediated decay are also able to contribute to transcriptional regulation and thus can serve as important components of gene expression regulatory networks. Furthermore, given our frequent use of the term in the manuscript, substituting with the term “pseudo-mRNA” with “aberrantly spliced mRNA” throughout would likely be more cumbersome for the reader. However, we have now defined in the abstract the term pseudo-mRNA as first coined by Frith et al (PLoS genetics 2006).

We do find the term “symmetric” and “asymmetric” exons to be used in literature relating to the topic of exon shuffling and to be also useful here in our manuscript. However, we agree with the reviewer these terms can be better defined. We have now done so in the Results section where labels are first introduced “Exon phase symmetry influences CRISPR/Cas9 outcomes...”:

We now include by way of introduction to the exon symmetry: “We noted in both these cases that the exons were symmetric – meaning the exon harbors a nucleotide number in multiples of three, and exclusion of this exon would result in a transcript that retains the original reading frame”.

The analysis of the role of secondary suggests possibilities (“may facilitate ATI”, “may cause stalling”) but a role for secondary structure is not really demonstrated by mutations that disrupt then re-establish to determine whether there are the predicted effects.

Response: We admit to the limitations of our in silico modeling approach to understanding the role of secondary structure in alternative translation initiation (ATI). We have moved the computational prediction of RNA changes induced by INDELS to the supplement and relegated to the discussion section our thoughts on this hypothesis. Some additional caveats are also offered with respect to this model (see second to last paragraph of DISCUSSION).

As noted above there are issues of clarity in the text and more direct simple statements would be helpful. For example, the main point for this is not clear – their presentation of the effect of the 1 bp insertion and why is hard to figure out. “We noted that a cDNA harboring the 1 bp insertion that provoked the ATI LKB1 protein was not associated with a redistribution of the initiation site suggesting leaky scanning is not likely responsible for the ATI event (Supp. Fig. 3). Given that the PTC is located 3’ to the predicted ATI site, we also assume this is not a re-initiation phenomenon in which translational termination is followed by the ribosome re-launching translation at a secondary AUG codon.” Also what specifically is “pressure-tested” rather than just tested?

Response: We apologize for lapses in clarity in our presentation. We have (hopefully) simplified our discussion of this data and made similar modifications throughout the

manuscript to improve our communication (ie. “tested” instead of “pressure-tested”, and simplification of discussion regarding ATI mechanisms as noted above)

The test of the CRISPinatoR design tool was modest targeting only two exons. ESE and ESS prediction algorithms are imperfect and this should be described since the paper presents identification of these elements as a given.

Response: In our study, we have provided data that support a potential utility of this algorithm. However, we agree that offering more caveats/context to our discussion of ESE and ESS prediction algorithms would be helpful to the reader. We have also provided additional references that speak the reliability of the ESE/ESE sequence predictions for anticipating the effects of exonic mutations on splicing integrity (see Zatkova et al, and Baralle et al).

REVIEWERS' COMMENTS:

Reviewer #1 (Remarks to the Author):

The authors have addressed my concerns.

One final suggestion: that premature stop codons proximal to the normal initiation codon fail to trigger NMD because of downstream re-initiation was previously discovered through a large-scale analysis of cancer genomes (<https://www.nature.com/articles/ng.3664>). One of the authors' cases appears to be an example of this mechanism.

Response to Reviewer #1:

The authors have addressed my concerns.

One final suggestion: that premature stop codons proximal to the normal initiation codon fail to trigger NMD because of downstream re-initiation was previously discovered through a large-scale analysis of cancer genomes (<https://www.nature.com/articles/ng.3664>). One of the authors' cases appears to be an example of this mechanism.

Response: The suggested reference has now been added.